# CrystalTrace: A Monte Carlo Raytracing Algorithm for Radiative Transfer in Cirrus Clouds with Oriented Ice Crystals

Linda Forster[1,2], Anna Weber[1], and Bernhard Mayer[1,3]

[1]Meteorologisches Institut, Ludwig-Maximilians-Universität, München, Germany.
[2]now at Jet Propulsion Laboratory, California Institue of Technology, Pasadena. United States.
[3]also at Institut für Physik der Atmosphäre, Deutsches Zentrum für Luft- und Raumfahrt, Oberpfaffenhofen, Germany.

**Correspondence:** Linda Forster (linda.i.forster@jpl.nasa.gov)

**Abstract.** We present CrystalTrace, a Monte Carlo raytracing algorithm designed for radiative transfer computations involving ice crystals with both random and preferred orientations. Integrated into the three-dimensional radiative transfer solver MYSTIC, which is part of *libRadtran*, CrystalTrace enables multiple-scattering simulations of absolute radiances for realistic atmospheric scenes, including clouds with different mixtures of ice crystals and water droplets, aerosols, molecules, and surface properties. It seamlessly extends MYSTIC's macroscopic raytracing down to the microscopic scale of individual ice crystals. By computing single-scattering properties online, CrystalTrace removes the need for precomputed look-up tables, which are also subject to limited angular resolution. The current version of CrystalTrace is implemented for the visible spectral range and supports individual hexagonal prisms with different sizes and aspect ratios. It currently does not account for diffraction and ice absorption. CrystalTrace, integrated with MYSTIC, fills a critical gap by providing a computationally efficient forward radiative transfer simulator for ice clouds containing oriented crystals, thereby enabling retrievals of ice crystal properties from ground-based, airborne, and satellite imaging observations.

## 1 Introduction

Ice crystal orientation has significant effects on the global radiation budget (Noel and Sassen, 2005) and oriented crystals have been frequently observed by space-borne observations (Chepfer et al., 1999; Noel and Chepfer, 2004, 2010; Zhou et al., 2012; Saito et al., 2017). Ice crystal shape and orientation also determines the fall speed of crystals in ice clouds which strongly affects climate sensitivity in GCM (General Circulation Model) simulations (Sanderson et al., 2008; Mitchell et al., 2008). Better knowledge of ice crystal shape, surface roughness, and orientation are therefore essential for improving estimates of the radiative forcing of cirrus clouds as well as satellite retrievals of cirrus optical and microphysical properties (e.g. Yang et al. (2015); Liou and Yang (2016)). Remote sensing of cirrus cloud properties also depends crucially on an accurate representation of the ice crystal single scattering properties which are key to correctly model the enhanced backscattering from horizontally oriented ice crystal faces. In the solar spectrum these oriented crystal faces act as mirrors reflecting a significantly larger amount of radiation back to space compared to randomly oriented crystals.

From an airplane window or mountain top, cirrus clouds consisting of oriented crystals often form optical displays, such as a sub-sun, sub-sun parhelia, and a lower parhelic circle. In fact, frequently observed halo displays such as sundogs and upper

tangent arcs can only be explained by the presence of oriented ice crystals (e.g. Wegener, 1925; Tricker, 1970; Tape, 1994; Tape and Moilanen, 2006). Continuous HaloCam observations during the ACCEPT campaign revealed that the majority (73%) of observed halo displays were produced by oriented crystals (sometimes together with randomly oriented crystals) while the minority (27%) were produced by randomly oriented crystals only (Forster et al., 2017). An example of halo displays produced

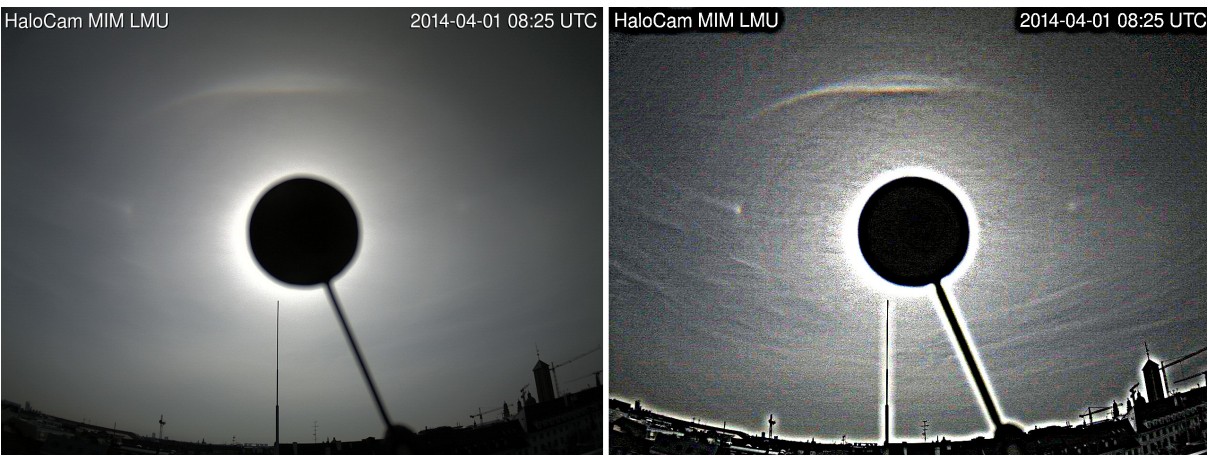

**Figure 1.** HaloCam$_{\text{JPG}}$ image from 1 April 2014 during the ML-CIRRUS campaign. Left: original HaloCam$_{\text{JPG}}$. Right: same image with an unsharp mask filter applied to better highlight the optical features.

by oriented ice crystals is shown in Fig. 1: An upper tangent arc and sundogs (22° parhelia) are visible, which are produced
by oriented columns and plates, respectively. Above the tangent arc, another arc is visible which had been present for several minutes. This is most likely a rare Parry arc which forms by ice crystal columns with a specific horizontal orientation. This halo type is named after W. E. Parry who first described this type of halo display between 1819 and 1820 (e.g. Pernter and Exner, 1910; Tricker, 1970; Greenler, 1980) and is extremely rare to observe.

Radiative transfer simulations usually assume that scattering particles are randomly oriented within the atmosphere. This
assumption greatly simplifies the parameterization of scattering because the phase function then depends only on the scattering angle. For liquid water clouds, random orientation is realistic. For ice crystals with preferred orientation, however, the single-scattering properties depend on the azimuth of the scattering plane as well as the zenith angle of the incident and outgoing ray (Liou and Yang, 2016). For a look-up-table-based approach, this implies storing the optical properties for each incident and scattered angle. Storing orientation-dependent phase matrices for all combinations of incident and scattered angles would
increase storage by roughly a factor of $10^7$ ($360\times180\times180$) for 1° resolution compared to randomly oriented crystals. A practical alternative is to compute radiances directly using a geometric raytracing method to calculate the radiance field caused by the refraction and reflection of sunlight by ice crystals. This method was first developed by Wendling et al. (1979) for hexagonal ice crystal columns and plates and later advanced by using more complex particles and including absorption and polarization effects (Pattloch and Tränkle, 1984; Muinonen et al., 1989; Takano and Liou, 1990; Macke, 1994; Takano and
Liou, 1995; Hess, 1996; Macke et al., 1996; Yang and Liou, 1998; Prigarin, 2009).

This study presents CrystalTrace, a computationally efficient raytracing model for ice crystals of different shapes with both random and preferred orientation. A key advantage of the geometric raytracing method is its compatibility with Monte Carlo methods which makes it straightforward to integrate into *libRadtran*'s MYSTIC solver to allow simulating absolute radiances under realistic atmospheric conditions. This manuscript is organized as follows: Section 2 introduces CrystalTrace, Section 2.1 describes its implementation, and Section 2.2 its integration into MYSTIC. Section 2.3 presents CrystalTrace's validation, followed by a demonstration to simulate different halo displays in Section 2.4 including a real-world example.

## 2 CrystalTrace – the Monte Carlo Raytracing Algorithm

CrystalTrace is a Monte Carlo raytracing algorithm for simulating light scattering by non-spherical ice crystals in the geometric-optics limit. It supports mixtures of different ice crystal shapes with varying orientation angles, along different symmetry axes. The raytracing technique is computationally fast compared to rigorous electromagnetic methods like T-matrix or Discrete-Dipole Approximation (Draine and Flatau, 1994; Flatau and Draine, 2014) which is an important advantage for the use of radiative transfer simulations. We focus on the visible spectral range, where the wavelength is much smaller than typical ice crystal sizes resulting in size parameters of the order of 100. Raytracing is applicable when the particle size is much larger than the wavelength, i.e. for a size parameter $\chi \gg 1$ (Liou and Yang, 2016). In this regime, incident light can be approximated as a bundle of localized rays undergoing external and internal reflections and refractions. Furthermore, it has to be assumed that the energy attenuated by the scatterer may be decomposed into equal extinction from diffraction and from Fresnel rays. The geometric optics approximation can be applied to crystals with particle sizes large compared to the wavelength of the incident radiation. For radiative transfer simulations of oriented ice crystals in the visible spectrum, this is well justified since the tendency of crystals to assume preferred orientations increases with size. According to Konoshonkin et al. (2017) the accuracy of the results will be at least 95% for crystals with size parameters of 120 and larger.

### 2.1 Implementation of CrystalTrace

The implementation of CrystalTrace will be explained in the following which is building on the methodology described by Macke (1994). The current version of CrystalTrace does not account for the polarization state, diffraction, ice absorption and ice crystal surface roughness which could be incorporated in the future. The algorithm starts with:

**Step 1 – Select a crystal shape.** The default ice crystal shape implemented in CrystalTrace has a hexagonal (six-fold) symmetry. Hexagonal crystals are typical for the Earth's atmosphere and result from the non-uniform deposition coefficients across the crystal surface (e.g. Magono and Lee (1966)). The majority of halo displays observed on Earth can also be traced back to hexagonal prisms. CrystalTrace provides hexagonal prisms as shown in Fig. 2 with different aspect ratios ranging from columns (a) to plates (b), all of which are composed of two basal faces and six prism faces. The length of the $c$-axis is referred to by $L$, while the maximum diameter of the ice crystal top and base faces along the $a$-axis is denoted by $2a$. A dedicated Python tool is provided with the CrystalTrace GitHub repository, which takes any combination of $L$ and $a$ (both in units $\mu m$)

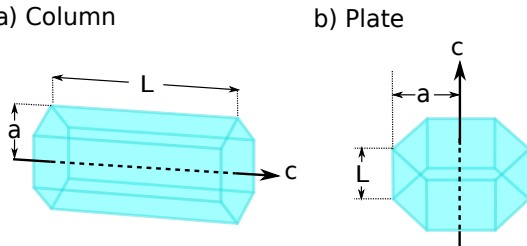

**Figure 2.** Definition of ice crystal symmetry axes and dimensions. The length of the $c$-axis is denoted by $L$. The maximum diameter of the ice crystal top and base is defined by $2a$ along the $a$-axis.

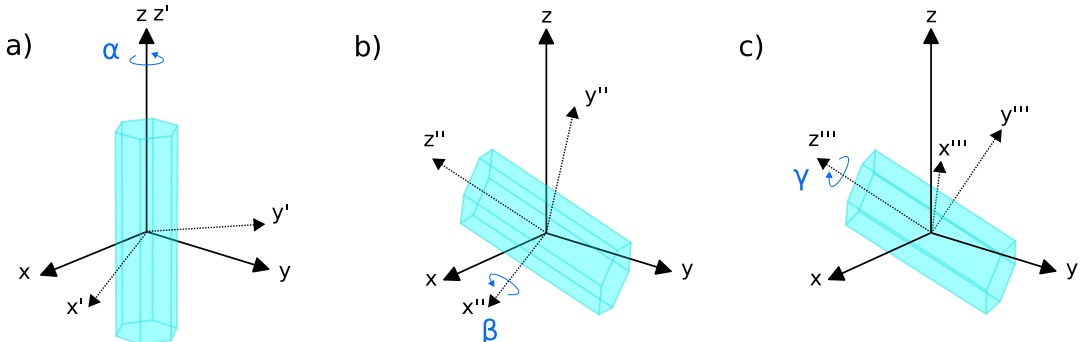

**Figure 3.** Ice crystal geometry and definition of Euler angles as used for CrystalTrace. The reference coordinate system is represented by the $x$, $y$, and $z$-axes. (a) Original geometry of the ice crystal rotated by the Euler angle $\alpha_\text{Euler}$ around its $z$-axis. The axes of the updated coordinate system are denoted by $x'$, $y'$, and $z'$. The second rotation by $\beta_\text{Euler}$ around the $x'$-axis results in the crystal orientation shown in (b) with the updated coordinate system denoted by $x''$, $y''$, and $z''$. (c) Third rotation around the $z''$-axis by $\gamma_\text{Euler}$.

as input to compute the geometry of a corresponding hexagonal prism. Since diffraction is not implemented yet, note that differences in ice crystal size will have no effect on the result as long as the aspect ratio remains constant.

**Step 2 – Determine Ice Crystal Orientation.** The crystal's orientation is calculated by randomly sampling three Euler angles $\alpha_\text{Euler}$ (angle of crystal c-axis), $\beta_\text{Euler}$ and $\gamma_\text{Euler}$. The Euler rotations around these three angles are illustrated in Fig. 3. Three configurations with different degrees of freedom (DOF) for the choice of Euler angles are usually distinguished:

1. **Randomly oriented crystals**: for randomly oriented crystals the new orientation is determined by drawing random numbers $\varrho$ between $[0, 1)$ for each of the three Euler angles

$$\alpha_\text{Euler} = 2\pi\,\varrho\,, \tag{1}$$

$$\beta_\text{Euler} = \arccos(1 - 2\varrho)\,, \tag{2}$$

$$\gamma_\text{Euler} = 2\pi\,\varrho\,, \tag{3}$$

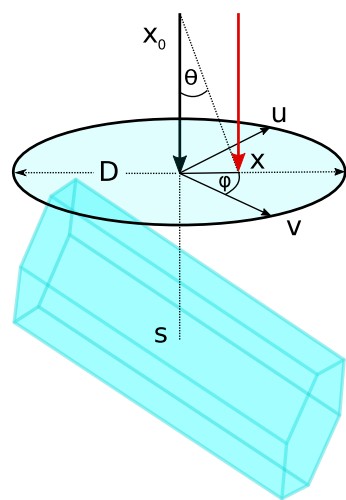

**Figure 4.** Method to determine the initial point $\mathbf{x}$ of the incident ray $\mathbf{x_0}$. The red arrow denotes the incident direction of the ray with starting point $\mathbf{x}$. The intersection point with the ice crystal $\mathbf{x}$ is determined by drawing random angles $\theta$ and $\phi$ in a circle centered around the barycenter $s$ with diameter $D$, the maximum dimension of the ice crystal. The vectors $\mathbf{u}$ and $\mathbf{v}$ are auxiliary vectors which are perpendicular to $\mathbf{x}$ and to each other.

which corresponds to a $\mathrm{DOF}=3$. This configuration of ice crystal orientation is responsible for formation of the 22 and 46° halo.

2. **Singly oriented ice crystals**: Singly oriented ice crystals cause the upper and lower tangent arcs in case of columns and sundogs in case of plates. These ice crystals with preferred orientation of their $c$-axis are implemented by sampling the Euler angles from a Gaussian distribution with mean value $\mu_{\beta,\mathrm{Euler}}$ in the direction of orientation and a standard deviation $\sigma_{\beta,\mathrm{Euler}}$ to represent oscillations of the $c$-axis around the preferred orientation:

$$
\beta_{\mathrm{Euler}} = \frac{1}{\sqrt{2\sigma_{\beta,\mathrm{Euler}}^2\,\pi}} \exp\left(-\frac{(\varrho - \mu_{\beta,\mathrm{Euler}})^2}{2\,\sigma_{\beta,\mathrm{Euler}}^2}\right). \tag{4}
$$

The standard deviation of the Gaussian orientation distribution is referred to as *orientation parameter*. Ice crystal plates have aerodynamically stable orientations with their $c$-axis vertical (similar to leaves falling from trees) which implies $\mu_{\beta,\mathrm{Euler}} = 0°$. Ice crystal columns instead tend to orient their $c$-axis horizontally and thus $\mu_{\beta,\mathrm{Euler}} = 90°$. This configuration has $\mathrm{DOF}=2$ and is referred to as singly oriented since only one crystal axis is oriented and the other two Euler angles $\alpha_{\mathrm{Euler}}$ and $\gamma_{\mathrm{Euler}}$ are chosen randomly as in (1).

3. **Parry oriented ice crystals**: in rare cases horizontally oriented ice crystal columns can also be oriented in their rotation around $z'''$ (cf. Fig. 3c) which reduces their DOF to 1. This means that both $\beta_{\mathrm{Euler}}$ and $\gamma_{\mathrm{Euler}}$ are parameterized by a Gaussian distribution in analogy to Eq. 4. Only $\alpha_{\mathrm{Euler}}$ is sampled from a uniform distribution between 0 and $2\pi$. These oriented ice crystal columns form the rare Parry arc which is visible above the upper tangent arc in Fig. 1.

**Step 3 – Intersect Light Ray and Crystal Surface.** Each ray is described by the intersection of two plane equations with normal vectors corresponding to the oscillating plane of the electric field vectors. The intersection of these two planes with the plane of the crystal provides the point at which the ray hits the crystal surface. The point of intersection between the direction of the incident ray $\mathbf{x_0}$ and the plane of a crystal face is determined at a randomly chosen point. This procedure is illustrated in Fig. 4: a circle is defined in the plane perpendicular to $\mathbf{x_0}$ and centered around the geometric center of the ice crystal $\mathbf{s}$. The radius is determined by the maximum diameter $D$ of the ice crystal. Within this circle a point is chosen by drawing random numbers for the zenith ($\theta$) and azimuth angle ($\phi$). The resulting vector of the incident ray is denoted by a red arrow in Fig. 4 and is calculated by

$$\mathbf{x} = \mathbf{s} - \mathbf{x_0} \cdot D/2 + (\cos\phi\,\mathbf{u} + \sin\phi\,\mathbf{v})\,\sqrt{\theta} \cdot D/2\,. \tag{5}$$

Random points are repeatedly selected until an intersection is found between the plane of an ice crystal face and the vector of the incident ray $\mathbf{x}$.

**Step 4 – Reflection or Transmission?** Once a point of intersection with the crystal surface is found, the ray is either reflected or transmitted. Fresnel's equations quantify the intensity of the reflected and transmitted light. Those equations are derived by decomposing the electric fields into their components parallel $E_{r\parallel}$ and perpendicular to the plane of incidence $E_{r\perp}$ and applying the respective continuity conditions to obtain the reflection coefficients of the field components parallel ($r_\parallel$) and perpendicular ($r_\perp$) relative to the plane of incidence:

$$r_\perp = \frac{E_{r\perp}}{E_{i\perp}} = -\frac{\sin(\theta_i - \theta_t)}{\sin(\theta_i + \theta_t)}\,, \tag{6}$$

$$r_\parallel = \frac{E_{r\parallel}}{E_{i\parallel}} = \frac{\tan(\theta_i - \theta_t)}{\tan(\theta_i + \theta_t)}\,. \tag{7}$$

The reflection coefficient is computed by $\mathcal{R} = (r_\parallel^2 + r_\perp^2)/2$ and the transmission coefficient by $\mathcal{T} = 1 - \mathcal{R}$. To determine whether the ray is reflected or transmitted, a random number $\varrho$ is sampled uniformly within $[0, 1)$. If $\varrho < \mathcal{R}$, the ray is reflected, otherwise it is transmitted. Total reflection occurs if the angle of refraction is $\theta_t > \arcsin(1/n_{\mathrm{Re}})$. In this case, the probability for reflection is 100% and the ray is externally reflected. The ray paths for refraction and reflection at a hexagonal ice crystal is illustrated in Fig. 5. The scattering geometry is equivalent to a wedge with opening angle $\Delta$. The direction of the reflected and refracted rays can be calculated using Snell's law

$$n_i \sin\theta_i = n_t \sin\theta_t\,. \tag{8}$$

Snell's law describes the relation between the angle of incidence $\theta_i$ in a medium with refractive index $n_i$ and the angle of refraction $\theta_t$ in a medium with refractive index $n_t$. The refractive index is used from Warren Wiscombe's REFICE code using ice refractive indices from Warren (1984). With the known incident angle $\theta_i$ Eq. 8 can be solved for the transmitted angle $\theta_t$. In case of reflection, the refractive indices are the same and cancel in Eq. 8 which reduces to $\theta_i = \theta_r$, i.e. the angle of reflection is equal to the angle of incidence.

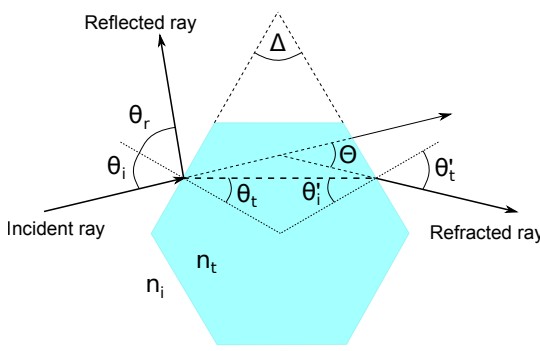

**Figure 5.** Refraction and reflection ray paths illustrating the scattering geometry for Snell's law. $\Delta$ is the opening angle of the prism, $\theta_i$ is the angle of incidence, $\theta_r$ the angle of reflection, and $\theta_t$ the transmission angle. $\theta'$ denotes the angles of the second incident and transmitted ray, and $\Theta$ is the scattering angle between the incident ray entering the crystal and the refracted ray exiting the crystal. $n$ denotes the refractive index of the incident medium $n_i$ (here air, $n_i \approx 1$) and the transmission medium $n_t$ (here ice, $n_t \approx 1.31$).

After the first transmission, ray tracing is continued inside the ice crystal as long as the ray is reflected internally off the crystal surfaces until it gets transmitted again and exits the crystal. Once the ray exits the crystal or in case of initial external reflection (the ray never enters the crystal), the CrystalTrace routine ends and the ray gets counted in the respective ($\mu$, $\phi$) bin at the top or bottom of the domain depending on the sign of the vector's $z$ component.

## 2.2 Integrating CrystalTrace in the Monte Carlo Radiative Transfer Model MYSTIC

MYSTIC solves the radiative transfer equation with the Monte Carlo method by tracing photons through the atmosphere. For radiative simulations of ice clouds, MYSTIC uses a look-up table approach which determines the outgoing angle of the light ray for a given incident angle by randomly sampling the scattering angle from the phase function (cf. Fig. 6). For randomly oriented ice crystals, this is an effective approach since the relative azimuth angle of the outgoing ray can be sampled randomly and is independent of the scattering angle. For crystals with preferred orientation however, the scattering angle depends both on the zenith and azimuth angles relative to the plane of scattering, which causes a challenge on computing memory as well as interpolation techniques for the look-up table approach. With CrystalTrace implemented in MYSTIC, the raytracing simply continues inside the crystal once a scattering event takes place and the direction of the outgoing ray directly results from tracing the ray through the crystal. The scattering properties are effectively computed online during the simulations which allows for radiative transfer simulations without explicit knowledge of the scattering phase function. Another advantage is that no delta-scaling is needed since photons refracted by two parallel crystal faces escape the crystal and are traced further through the atmosphere.

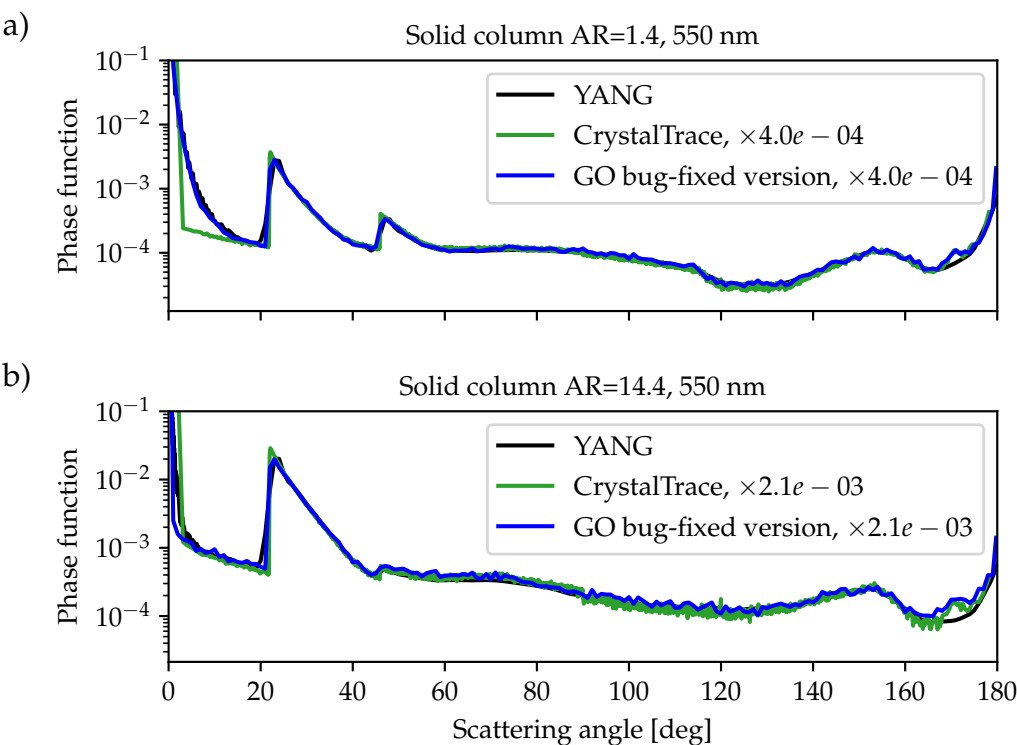

**Figure 6.** Comparison of the scattering phase functions calculated with YANG (black), CrystalTrace (green) and GO (blue) at wavelength of 550 nm. The GO results here were computed with the bugfixed version of Macke (1994) as described in Forster et al. (2017) (see additional details in the Supplement). (a) Scattering phase functions for a solid ice crystal column with $AR = 1.4$ and maximum dimension $D = 65\,\mu m$ and (b) for $AR = 14.4$ with $D = 10^4\,\mu m$. Both the GO and CrystalTrace phase functions were scaled to allow for a comparison of the results for scattering angles excluding the forward scattering peak. The scaling factors are shown in the legend. The results compare well apart from the forward scattering region for the smaller ice crystal (a) due to diffraction which is not accounted for by CrystalTrace.

## 2.3 Validation of CrystalTrace

CrystalTrace was validated for both single- and multi-scattering simulations. The single-scattering simulations allow comparison with scattering phase functions from Macke's raytracing code (referred to by GO) and the database provided by Yang et al. (2013) (referred to by YANG) for ice crystals of different aspect ratios. Figure 6 compares the phase functions calculated with CrystalTrace (green) and GO (blue) for ice crystals columns with aspect ratio 1.4 (a) and long solid columns with aspect ratio 14.4 (b) at a wavelength of 550 nm. The corresponding phase function of the YANG database (black) is shown as a reference. The results compare well apart from the forward scattering region where CrystalTrace shows smaller values than GO and YANG. This is due to diffraction which is not accounted for by CrystalTrace. The difference is more pronounced for

small crystals (Fig. 6a) compared to large crystals (Fig. 6b). Due the coarser scattering angle resolution in the case of the GO phase functions the peaks of the 22 and 46° halos are not as pronounced as for CrystalTrace.

In addition to single scattering, CrystalTrace was validated for multi-scattering radiance simulations as well. As part of *libRadtran*'s MYSTIC solver, CrystalTrace allows simulations ice clouds with different aspect ratios and preferred or random orientations in a realistic atmosphere with air molecules and aerosol particles. To validate the simulated radiances, the results of MYSTIC with CrystalTrace were compared to MYSTIC with tabulated YANG optical properties, assuming randomly oriented ice crystals with the same size and aspect ratio. Figure 7 shows the radiance distribution for an ice cloud of optical thickness of 1, simulated with MYSTIC using (a) CrystalTrace and (b) YANG to solve for the ice crystal single scattering properties. Figure 7c represents the difference between the two simulations. Panels (d)–(f) show the standard deviation, representing the Monte Carlo noise within a 1-$\sigma$ confidence interval, which is on average about 9% relative to the simulated radiances. The mean difference in Fig. 7c is (0.0004 $\pm$ 0.0573) W m$^{-2}$ nm$^{-1}$ sr$^{-1}$, effectively zero within the 1-$\sigma$ confidence range. The bottom panel displays cross sections through the principal (g) and almucantar (h) planes.

As in Fig. 6, the largest differences occur near the forward peak due to diffraction. Smaller differences are found at the 22° halo peak, which appears sharper in the CrystalTrace results than in YANG. It is important to note that the tabulated scattering phase functions provided by YANG have a resolution of 1° in the scattering angle range, which is not fine enough to resolve the halo peak and therefore smooths out this scattering feature in the radiance distribution. This highlights an important advantage of computing single-scattering properties online with CrystalTrace: the results are not constrained by the angular resolution of a precomputed look-up table. The lower YANG radiances at the 22° halo peak arise from two factors: (1) the coarse sampling of the scattering angle grid of YANG which limits the maximum resolvable peak height and (2) neglecting diffraction may enhance radiances in the CrystalTrace simulations. In addition to radiances, we compared top-of-atmosphere (TOA) and bottom-of-atmosphere (BOA) fluxes for randomly oriented crystals by integrating radiances over the respective hemispheres. For all configurations, the net fluxes are effectively zero within the 1-$\sigma$ uncertainty, confirming energy closure.

**Table 1.** Radiative flux simulations at top-of-atmosphere (TOA) and bottom-of-atmosphere (BOA) with MYSTIC, comparing CrystalTrace and YANG for a solar zenith angle of $\theta_0 = 30°$ and azimuth angle $\phi_0 = 270°$, columns with AR = 14.4, D = $10^4$ $\mu$m at a wavelength of 550 nm, and optical depths $\in [1, 10]$. The simulations were performed without atmospheric background or ice absorption. The net flux is defined as $E_{\text{net}} = E_{\text{dir,TOA}} - E_{\text{dir,BOA}} - E_{\text{dn,BOA}} - E_{\text{up,TOA}}$. Provided are the average and standard deviation over $10^7$ photons. The uncertainty represents a 1$\sigma$-confidence interval.

| ic_properties | $\tau$ | $E_{\text{dir,TOA}}$ | $E_{\text{dir,BOA}}$ | $E_{\text{dn,BOA}}$ | $E_{\text{up, TOA}}$ | $E_{\text{net}}$ [mW m$^{-2}$ nm$^{-1}$ sr$^{-1}$] |
|---|---|---|---|---|---|---|
| YANG | 1 | 1444.72 | 391.50 $\pm$ 0.74 | 966.74 $\pm$ 0.70 | 86.43 $\pm$ 0.27 | 0.05 $\pm$ 1.06 |
| YANG | 10 | 1444.72 | 0.01 $\pm$ 0.00 | 816.67 $\pm$ 0.83 | 628.15 $\pm$ 0.62 | -0.10 $\pm$ 1.04 |
| CrystalTrace | 1 | 1444.72 | 391.36 $\pm$ 0.46 | 922.08 $\pm$ 0.66 | 131.39 $\pm$ 0.40 | -0.10 $\pm$ 0.90 |
| CrystalTrace | 10 | 1444.72 | 0.00 $\pm$ 0.00 | 676.31 $\pm$ 0.75 | 768.26 $\pm$ 0.92 | 0.15 $\pm$ 1.19 |

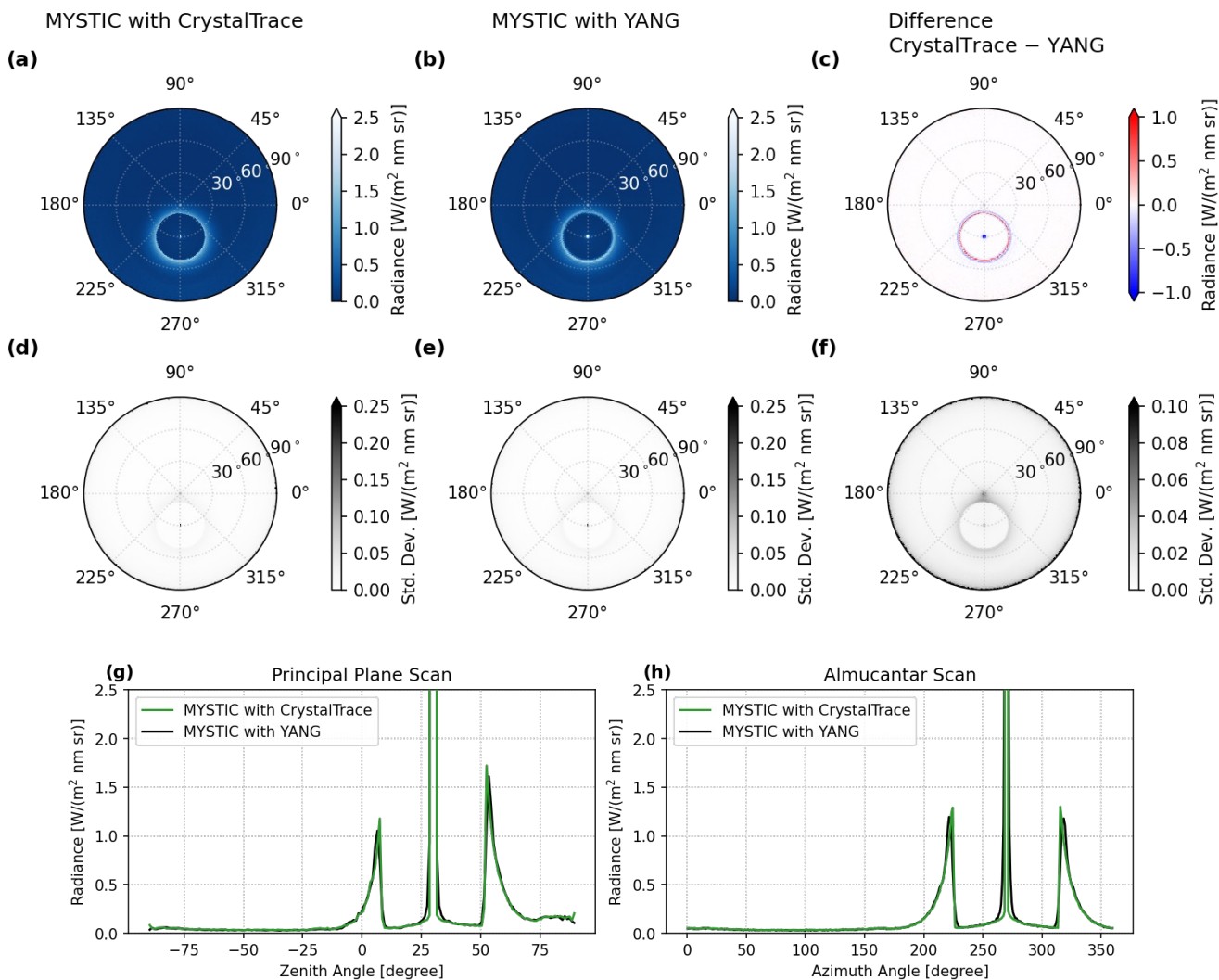

**Figure 7.** Comparison of simulated radiances for a cirrus cloud of optical thickness 1 with randomly oriented ice crystals using optical properties of the YANG database and CrystalTrace for $AR = 14.4$ with $D = 10^4\,\mu m$ at a wavelength of 550 nm. The simulations were performed with $4 \times 10^7$ photons at a solar zenith angle of $\theta_0 = 30°$ and azimuth angle $\phi_0 = 270°$, without atmospheric background. The top panel shows a polar projection of the simulated radiances with MYSTIC using (a) CrystalTrace and (b) the YANG look-up table to solve for the ice crystal single scattering properties. Panel (c) shows the difference between CrystalTrace and YANG and (d) – (f) show the corresponding standard deviation. The bottom panel displays cross sections through the polar plots along the principal (zenith angle transect along $\phi = \phi_0$) (g) and almucantar plane (azimuth angle transect along $\theta = \theta_0$) (h). Here, the MYSTIC radiances are computed using CrystalTrace are indicated by the green line and YANG in black.

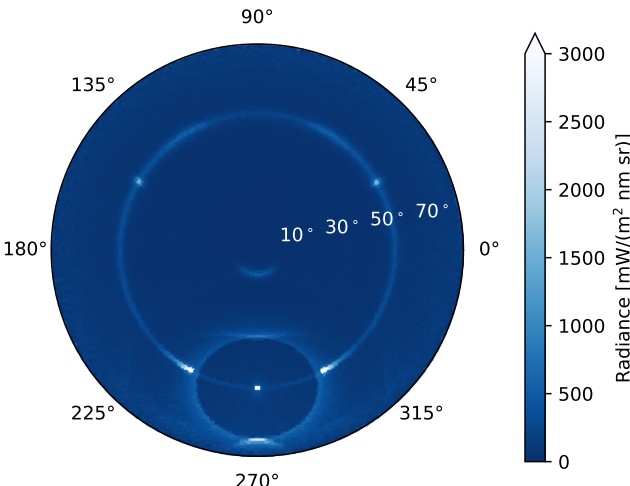

**Figure 8.** Raytracing simulation with MYSTIC and CrystalTrace showing the radiance distribution in a polar projection for the upper hemisphere using $10^7$ photons. The zenith angle $\theta$ ranges from 0 to $90°$ at the horizon, whereas the azimuth angle $\phi$ ranges from 0 to $360°$. The sun is located at an SZA of $60°$ and an azimuth angle of $\phi_0 = 270°$. The simulation was performed for a wavelength of 550 nm and an ice cloud of optical thickness of 0.8. The cirrus cloud contains 40% oriented ice crystal columns (c-axis horizontal), 40% oriented ice crystal plates (c-axis vertical), and 20% randomly oriented ice crystal columns. For both oriented crystal populations an orientation parameter of $\sigma_{\beta,\mathrm{Euler}} = 1°$ was chosen.

## 2.4 Application of CrystalTrace

To demonstrate the capabilities of CrystalTrace implemented in MYSTIC, we showcase simulations of cirrus clouds containing mixtures of crystals with different orientations and aspect ratios under realistic atmospheric and solar illumination conditions. Figure 8 shows the radiance distribution simulated with MYSTIC and CrystalTrace for a cirrus with optical thickness 0.8 in an aerosol-free atmosphere with non-reflective surface. A solar zenith angle of $60°$ and an azimuth angle of $270°$ were selected. The cirrus cloud contains 40% oriented ice crystal columns (c-axis horizontal), 40% oriented ice crystal plates (c-axis vertical), and 20% randomly oriented ice crystal columns. For both oriented crystal populations an orientation parameter of $\sigma_{\beta,\mathrm{Euler}} = 1°$ was chosen. The oriented plates are responsible for the sundogs ($22°$ parhelia) visible on each side of the sun, the $120°$ parhelia, and the circumzenithal arc, which is visible at $\theta \approx 10°$. All parhelia are located on the parhelic circle which forms at the zenith angle of the sun $\theta = \theta_0$ and is produced by reflection at vertical crystal faces of both oriented plates and columns. The oriented hexagonal columns produce the upper and lower tangent arcs visible above and below the sun in Fig. 8 and the $22°$ halo forms due to the presence of randomly oriented hexagons. This simulation was performed using $10^7$ photons and $360 \times 180$ ($\phi \times \theta$) bins, resulting in about 154 photons per bin. With CrystalTrace, MYSTIC has to be used in the "forward tracing mode" (from the source to the detector), which has a similar computational time as the backward mode (from the detector to the source) if

both hemispheres are simulated. The runtime for $10^7$ photons amounts to about 50 s for $\tau = 1$ (140 s for $\tau = 10$) using a single core Apple M1 Max processor with 3.2 GHz clock frequency and is the same for CrystalTrace and YANG.

### 2.4.1 Oriented Ice Crystal Plates

Due to their preferred orientations, the scattering geometry for ice crystals forming sundogs as well as upper and lower tangent arcs strongly depends on the solar elevation. Figure 9 shows CrystalTrace simulations with oriented ice crystal plates for a



**Figure 9.** Sundogs for increasing solar elevations, calculated with CrystalTrace.

range of solar elevations starting close to the horizon at 20° to 60°. The raypath for the sundogs is similar to the 22° halo, but only for a solar elevation of 0°, i.e. at sunrise or sunset, when the plane of incidence is perpendicular to the $c$-axis. For larger

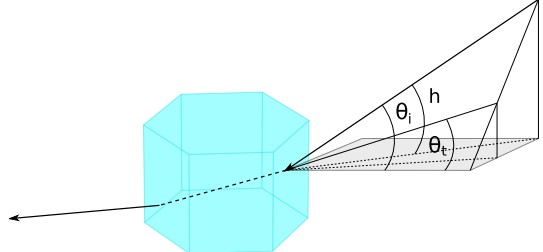

**Figure 10.** Geometry of a skew ray producing a sundog in an ice crystal plate. The solar elevation angle is denoted with $h$. The incident and transmitted angles are indicated by $\theta_i$ and $\theta_t$, respectively.

solar elevations the raypath is skewed due to the tilted plane of incidence relative to the crystal $c$-axis, as shown in Fig. 10. For solar elevations larger than about 60° the light rays cannot follow the path of minimum deviation since the scattering plane is not perpendicular to the $c$-axis. This is in agreement with the CrystalTrace simulations in Figure 9 at 60° solar elevation when

the sundogs start to vanish. This cut-off angle can be computed analytically as shown in Minnaert (1993) and Forster (2017).

The Supplement shows additional CrystalTrace simulations for ice crystals mixtures of randomly and preferred orientations. The transition of the sundogs to a 22° halo by gradually increasing the orientation parameter of the ice crystal plates is shown in Fig. S3 of the Supplement.

### 2.4.2 Oriented Ice Crystal Columns

Upper and lower tangent arcs are caused by ice oriented crystal columns with their $c$-axis horizontal. Light is refracted through two crystal side faces, similar as for the 22° halo and the sundogs. These arcs are located tangential to the 22° halo (cf. Fig. 11). The shape of the upper and lower tangent arc depends strongly on the solar elevation, as illustrated in Fig. 11 using raytracing

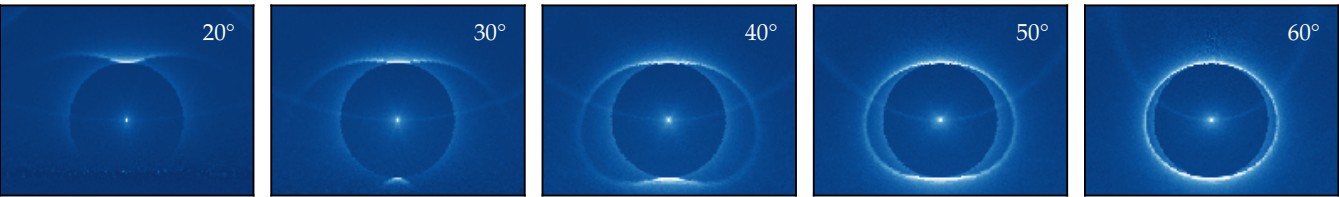

**Figure 11.** Upper and lower tangent arc merging into circumscribed halo for increasing solar elevations, calculated with CrystalTrace.

simulations. For solar elevations close to the horizon up to about 30°, the upper and lower tangent arcs are separated. As the solar elevation further increases towards the zenith, the wings of the upper and lower tangent arcs are getting closer until, at solar elevations larger than about 30°, they start to merge and form the circumscribed halo. For high solar elevations the circumscribed halo approaches the 22° halo until the two halos coincide for the sun at the zenith. CrystalTrace simulations for ice crystals mixtures of randomly and preferred orientations are shown in the Supplement.

### 2.4.3 Simulation of a Complex Halo Display during the ML-CIRRUS Field Experiment

On 1 April 2014, during the ML-CIRRUS campaign (Voigt et al., 2017) HaloCam$_{JPG}$ (Forster et al., 2017) recorded halo displays which formed in a very homogeneous cirrus layer over Munich and lasted for about 4 h. The halo displays showed a 22° halo simultaneously with an upper tangent arc and sundogs. High aerosol concentrations due to transported Saharan dust were reported during this period (Voigt et al., 2017). At 8:25 UTC an additional arc was visible on top of the upper tangent arc in Fig. 1, most likely a Parry arc originating from oriented ice crystal columns which have only 1 degree of freedom (see Section 2.1 Step 2c). We use CrystalTrace to reproduce this complex and rare halo display and confirm whether this additional arc stems from so-called Parry-oriented crystals. The results of the radiative transfer simulations is shown in Fig. 12 which are performed for a SZA of 57° at a wavelength of 500 nm. The Saharan dust layer is represented by an aerosol layer using OPAC's desert dust mixture (Hess et al., 1998) with an optical thickness of 0.48 (at 500 nm) as provided by the AERONET daily average. The simulation was performed for a COT of 1 and an ice crystal mixture of 45% randomly oriented columns which form the 22° halo, 10% oriented plates which cause the sundogs, 35% oriented columns which are responsible for the upper and lower tangent arcs, and 10% Parry oriented columns which form the upper Parry arc. The different ice crystal components were estimated to qualitatively reproduce the different halo displays visible in Fig. 1. As in Fig. 1, the Parry arc below the sun in Fig. 12 is not visible due to the large AOT and the long path through the atmosphere due to the low viewing zenith angles.

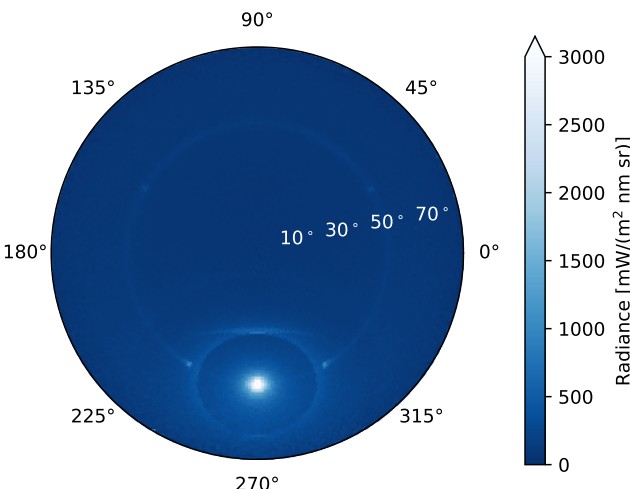

**Figure 12.** CrystalTrace simulation of 1 April 2014 8:25 UTC at an SZA of $57°$ and a wavelength of 500 nm. The simulation was performed for 45% randomly oriented columns which form the $22°$ halo, 10% oriented plates which cause the sundogs, 35% oriented columns which are responsible for the upper and lower tangent arcs, and 10% Parry-oriented columns which form the upper Parry arc. The simulation was performed for $10^7$ photons. A cirrus optical thickness of 1 was chosen and the aerosol was parameterized by the OPAC desert dust mixture with an AOT of 0.48 at 500 nm, which is the daily mean provided by AERONET.

Compared to Fig. 8, which was simulated for an aerosol-free atmosphere, Fig. 12 shows a strong forward scattering signal in the solar aureole region and the brightness of the halo displays is significantly reduced.

## 3 Summary and Conclusions

This study presents CrystalTrace, a Monte Carlo raytracing algorithm that simulates light scattering by non-spherical ice crystals in the visible spectrum, accounting for ice crystal orientation. As stand-alone tool, CrystalTrace can model single scattering properties for random as well as preferred ice crystal orientations with varying degrees of orientation, along different crystal symmetry axes. Integrated into the MYSTIC radiative transfer model, CrystalTrace provides an improved framework for simulating multi-scattering radiative transfer in cirrus clouds, in particular optical phenomena produced by oriented ice crystals, such as sundogs and upper/lower tangent arcs which are frequently observed in the atmosphere. This represents a significant advancement over traditional models that assume randomly oriented crystals and rely on precomputed look-up tables.

The ability to simulate real-time single scattering properties of ice crystals allows solving radiative transfer in an efficient and accurate way, not limited by computer memory or pre-defined angular resolution of look-up tables. CrystalTrace dynam-

ically computes scattering properties during simulations, offering higher resolution and greater flexibility for various crystal orientations.

We validated the single scattering properties simulated by CrystalTrace against precomputed scattering phase functions from the Yang et al. (2013) database (YANG). Moreover, we validated the multi-scattering radiances computed by MYSTIC & CrystalTrace against MYSTIC & YANG assuming randomly oriented crystal. The application of CrystalTrace to HaloCam observations from the ML-CIRRUS campaign demonstrates its effectiveness in reproducing the observed halo displays caused by oriented ice crystals. This real-world validation highlights the practical utility of the model and its potential for improving remote sensing retrievals of ice cloud properties. While CrystalTrace marks a significant advancement of simulating realistic radiances of clouds with oriented ice particles efficiently, future evolutions of the model could benefit from implementing the effects of polarization, diffraction, and ice crystal surface roughness, depending on the application.

CrystalTrace, integrated with the radiative transfer model MYSTIC and *libRadtran*, has demonstrated its potential for enhancing remote sensing retrievals of ice cloud optical and microphysical properties by accounting for ice crystal orientation. Better representation of ice crystal orientation and scattering properties can lead to more accurate satellite retrievals of cirrus cloud properties and improved estimates of cloud radiative forcing. Future developments in this area will continue to bridge gaps in our understanding of ice cloud microphysics, radiative effects, and their role in the Earth's climate system.

*Code and data availability.* CrystalTrace is distributed as part of the *libRadtran* radiative transfer package (version 2.0.6 and later). Supplementary materials, including detailed documentation, example input files, and plotting scripts used in this work, are available on GitHub (https://github.com/lforster/CrystalTrace) and have been preserved on Zenodo (DOI: 10.5281/zenodo.17679332). The ML-CIRRUS data will be provided upon requests by the authors.

*Author contributions.* LF developed CrystalTrace, performed testing and validation, and prepared the manuscript. AW contributed by further testing and validating CrystalTrace. BM and LF implemented CrystalTrace into MYSTIC. CrystalTrace was developed as part of LF's doctoral thesis. BM secured the funding for the HaloCam project and supervised the doctoral thesis. BM provided valuable feedback on the development and implementation of CrystalTrace, as well as on the manuscript.

*Competing interests.* At least one of the (co-)authors is a member of the editorial board of Atmospheric Measurement Techniques.

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
