# Peer review of "CrystalTrace: A Monte Carlo Raytracing Algorithm for Radiative Transfer in Cirrus Clouds with Oriented Ice Crystals"

_EGUsphere, 2025_

## Referee Comment (RC1)

**Review of "CrystalTrace: A Monte Carlo Raytracing Algorithm for Radiative Transfer in Cirrus Clouds with Oriented Ice Crystals," by Forster et al. 2025**

Forster et al. present a new radiative transfer algorithm for cirrus clouds containing oriented ice crystals. This scheme is based on a raytracing method that can be coupled to the existing MYSTIC radiative transfer model. It handles both single and multiple scattering by oriented and randomly oriented crystals with higher angular resolution, thereby improving the simulation of sharp halos. The authors compare the method with existing data-base and pre-tabulated approaches for phase functions and radiance fields, and they show that it reproduces several optical phenomena observed during a recent field campaign.

Major comments:

1. Spectral scope and absorption
   CrystalTrace is currently limited to the visible spectral range and neglects ice absorption. Although the code successfully reproduces halo phenomena, its suitability for radiance-based applications such as fluxes and heating-rate profiles remains uncertain. Even without absorption, please compare top- and bottom-of-atmosphere fluxes with those from other models under a zero-absorption assumption to validate energy closure.
2. Crystal shapes and aggregation
   The manuscript claims that CrystalTrace supports multiple ice-crystal habits, yet only hexagonal prisms are demonstrated. Which additional shapes are implemented, what parameters (aspect ratio, bullet length, etc.) can be set, and how are aggregates treated?
3. Missing diffraction and polarization (future work)
   The authors state that diffraction and full-vector (polarized) radiative transfer will be added in a future release. Please estimate how omitting diffraction biases the asymmetry parameter (g) and top-of-atmosphere short-wave flux for thin ($\tau \approx 1$) and thick ($\tau \approx 10$) cirrus layers of different crystal sizes.

Minor comments:
1. Please indicate whether CrystalTrace can be configured for active-sensor geometries (e.g. zenith-pointing or spaceborne lidar).
2. For the benefit of prospective users, consider supplying a concise example input deck (MYSTIC / libRadtran format) that defines an ice cloud with user-selected aspect ratio and orientation parameters.
3. Include a benchmark case with only randomly oriented crystals, comparing CrystalTrace results to those from an established solver. This will demonstrate that the new code reproduces standard solutions when orientation effects are disabled.

4.  Clarify whether the current implementation can handle mixed-phase clouds that contain both ice and liquid water.

Specific comments:

p.1 l-14: Please quantify the impact of ice-crystal orientation on the radiation budget (e.g. $\Delta$SW CRE).

p.3 l-46: The code is said to build on Macke (1994). Please spell out the improvements and novel aspects relative to that work. Section 2.3 compares CrystalTrace with the bug-fixed version of GO; clarify the differences between the two algorithms.

p.3 l-47: If a broader wavelength range than 400–700 nm is simulated, will the results near 400 nm and 700 nm remain consistent with other algorithms?

p.7 l-144: Is photon termination (e.g., Russian-roulette) applied to prevent endless scattering in very thick ice clouds?

p.8 l-173: The sentence contains two instances of "with"; please remove one.

p.8 l-184: The authors show that CrystalTrace is unaffected by scattering-angle resolution. What is the computational-cost difference between CrystalTrace and the pre-computed look-up table?

p.9 Figure 6: The legends list "4.0e-04" and "2.1e-03", but the meaning of these values is not explained.

p.10 Figure 7c: The background (non-22°) radiance is slightly higher in CrystalTrace. Was ice absorption enabled in the MYSTIC-YANG control run? If not, what explains the difference?

p.10 l-200: Please provide the standard deviation of radiances for MYSTIC-CrystalTrace versus MYSTIC-YANG over several independent runs.

p.10 l-202: Runtime is reported for only $1 \times 10^5$ photons—insufficient for low noise in a hemispheric simulation. Please also give the runtime (and peak memory) for an equivalent MYSTIC-YANG run to substantiate the claimed efficiency.

p.13 l-233: How many photons were used to obtain adequate aerosol-scattering statistics?

---

## Author Comment (AC1)

**Reply to referee comment RC2**

We thank the referee for carefully reviewing the manuscript and for the valuable suggestions and comments. The reviewer's comments are highlighted in blue and the authors' response in black. *Changes to the manuscript are displayed in italic font*.

The manuscript introduces CrystalTrace, a Monte Carlo raytracing algorithm designed for simulating radiative transfer in cirrus clouds containing ice crystals with both random and preferred orientations in the visible spectral range. It runs as a standalone tool to compute scattering phase functions and relative intensities for hexagonal prisms of varying sizes and aspect ratios, while being integrated into the MYSTIC solver within the libRadtran library to enable multiple-scattering simulations of absolute radiances in complex atmospheric scenes. The algorithm uses geometric optics, accounts for crystal orientations via Euler angles and Gaussian distributions for tilt, and validates its single-scattering properties against established methods like IGOM, showing good agreement except in forward scattering regions where diffraction is not included. CrystalTrace extends MYSTIC's capabilities by computing single-scattering properties online, reducing memory needs and eliminating precomputed look-up tables, and demonstrates its utility through simulations of atmospheric optical phenomena such as sundogs, tangent arcs, and Parry arcs. Overall, the tool addresses a gap in efficient forward modeling for oriented ice crystals, facilitating improved retrievals of cirrus properties from ground-based, airborne, and satellite observations.

Overall, this is an excellent study: well-organized, clearly written, and easy to follow. The development of CrystalTrace represents a valuable advancement in modeling radiative transfer for oriented ice crystals, addressing a key limitation in traditional approaches that assume random orientations. As an integrated component of the widely used libRadtran package, it holds significant potential for applications in remote sensing, and atmospheric optics, particularly for simulating optical phenomena like halos and improving cirrus cloud retrievals. I recommend that the manuscript be accepted for publication after the authors address the following minor clarifications and revisions.

**Minor comments:**

**Comment 1:** Currently, CrystalTrace does not account for diffraction, which is a known limitation of pure geometric optics (GO) methods for forward scattering regions. Can the authors provide more details on how diffraction effects are handled (or approximated) in the radiative transfer simulations described in Sections 2.3 and 2.4? For instance, did you apply forward peak truncation to the phase functions in the GO and IGOM simulations to mitigate delta-function transmission issues?
**Response 1:** CrystalTrace is a purely geometric optics model, and diffraction is not accounted for. In the radiative transfer simulations described in Sections 2.3 and 2.4, forward diffraction peak is represented by rays that continue propagating forward through the crystal. Because the model does not produce a delta function like forward peak in the phase function, no forward peak truncation or delta scaling was applied.

**Comment 2:** In Figure 7, there is a small peak around the 170° scattering angle in both the GO and CrystalTrace phase functions, which appears to be absent in the YANG phase function. Could the authors explain the potential cause of this difference?
**Response 2:** CrystalTrace inherited this small peak for the original GO raytracing solution, which was shown in Macke et al. 1996 and most likely stems from high-order internal reflection rays.

**Comment 3:** How straightforward is it for users to implement new crystal shapes in CrystalTrace?

For example, rare halos like the 9° halo have been linked to irregular ice crystal shapes such as droxtals (Zhang et al., 2004, Appl. Opt., 43, 2490-2501). Would adding such a non-hexagonal or complex geometry require significant code modifications, or is the framework flexible enough for users to extend it with minimal effort? Providing a brief example or guidance in the manuscript could enhance its accessibility.

**Response 3:** In principle, CrystalTrace can support the same types of hexagonal ice crystal geometries that Macke 1994's raytracing code has been used with. We have only tested hexagonal prisms with different aspect ratios, but the framework can be extended to other geometries. Users can generate new shapes by modifying the Python script in the GitHub repository that creates the geometry input files.

(https://github.com/lforster/CrystalTrace/blob/main/crystal_geometry/generate_crystal_file.py). We would like to note that for small, randomly oriented particles such as droxtals, optical properties from electromagnetic scattering methods, such as Yang et al. (2013) are preferred. The strength of CrystalTrace comes in for simulations of larger crystals with preferred orientations. We modified the abstract to clarify this.

*Page 1, line 7: "The current version of CrystalTrace is implemented for the visible spectral range and supports individual hexagonal prisms with different sizes and aspect ratios."*

**Comment 4:** Can the authors clarify in which version of libRadtran CrystalTrace will be (or has been) incorporated? As of the latest public release (version 2.0.6, December 2024), there is no mention of it on the official website, so specifying the target version or branch (e.g., a development or upcoming release) would help potential users.

**Response 4:** Thank you for this suggestion. CrystalTrace is part of libRadtran version 2.0.6 and includes an example as test case but has not yet been documented in the user manual. We provide additional documentation and examples in a GitHub repository. We modified the code and data availability statement:

*Page 15, line 261: "CrystalTrace is distributed as part of the libRadtran radiative transfer package (version 2.0.6 and later). Supplementary materials, including detailed documentation, example input files, and plotting scripts used in this work, are available on GitHub (https://github.com/lforster/CrystalTrace) and have been preserved on Zenodo (DOI: 10.5281/zenodo.17679332)."*

---

## Author Comment (AC2)

**Reply to referee comment RC1**

We thank the referee for carefully reviewing the manuscript and for the valuable suggestions and comments. The referee's comments are highlighted in blue and the authors' response in black. *Changes to the manuscript are displayed in italic font.*

Forster et al. present a new radiative transfer algorithm for cirrus clouds containing oriented ice crystals. This scheme is based on a raytracing method that can be coupled to the existing MYSTIC radiative transfer model. It handles both single and multiple scattering by oriented and randomly oriented crystals with higher angular resolution, thereby improving the simulation of sharp halos. The authors compare the method with existing data-base and pre-tabulated approaches for phase functions and radiance fields, and they show that it reproduces several optical phenomena observed during a recent field campaign.

**Major comments:**

**Comment 1. Spectral scope and absorption**
CrystalTrace is currently limited to the visible spectral range and neglects ice absorption. Although the code successfully reproduces halo phenomena, its suitability for radiance-based applications such as fluxes and heating-rate profiles remains uncertain. Even without absorption, please compare top- and bottom-of-atmosphere fluxes with those from other models under a zero-absorption assumption to validate energy closure.

**Response 1**: We thank the reviewer for this suggestion and included a comparison between MYSTIC/YANG and MYSTIC/CrystalTrace for TOA and BOA fluxes, assuming zero-absorption. In addition to comparing radiances (Fig. 7), we included Table 1 to validate energy closure (E_net).
*Page 9, line 178: "In addition to radiances, we compared top-of-atmosphere (TOA) and bottom-of-atmosphere (BOA) fluxes for randomly oriented crystals by integrating radiances over the respective hemispheres. For all configurations, the net fluxes are effectively zero within the 1-σ uncertainty, confirming energy closure."*

**Table 1.** Radiative flux simulations at top-of-atmosphere (TOA) and bottom-of-atmosphere (BOA) with MYSTIC, comparing CrystalTrace and YANG for a solar zenith angle of $\theta_0 = 30°$ and azimuth angle $\phi_0 = 270°$, columns with AR = 14.4, D = $10^4\,\mu$m at a wavelength of 550 nm, and optical depths $\in [1, 10]$. The simulations were performed without atmospheric background or ice absorption. The net flux is defined as $E_{net} = E_{dir,TOA} - E_{dir,BOA} - E_{dn,BOA} - E_{up,TOA}$. Provided are the average and standard deviation over $10^7$ photons. The uncertainty represents a $1\sigma$-confidence interval.

| ic_properties | $\tau$ | $E_{dir,TOA}$ | $E_{dir,BOA}$ | $E_{dn,BOA}$ | $E_{up, TOA}$ | $E_{net}$ [mW m$^{-2}$ nm$^{-1}$ sr$^{-1}$] |
|---|---|---|---|---|---|---|
| YANG | 1 | 1444.72 | $391.50 \pm 0.74$ | $966.74 \pm 0.70$ | $86.43 \pm 0.27$ | $0.05 \pm 1.06$ |
| YANG | 10 | 1444.72 | $0.01 \pm 0.00$ | $816.67 \pm 0.83$ | $628.15 \pm 0.62$ | $-0.10 \pm 1.04$ |
| CrystalTrace | 1 | 1444.72 | $391.36 \pm 0.46$ | $922.08 \pm 0.66$ | $131.39 \pm 0.40$ | $-0.10 \pm 0.90$ |
| CrystalTrace | 10 | 1444.72 | $0.00 \pm 0.00$ | $676.31 \pm 0.75$ | $768.26 \pm 0.92$ | $0.15 \pm 1.19$ |

**Comment 2. Crystal shapes and aggregation**
The manuscript claims that CrystalTrace supports multiple ice-crystal habits, yet only hexagonal prisms are demonstrated. Which additional shapes are implemented, what parameters (aspect ratio, bullet length, etc.) can be set, and how are aggregates treated?

**Response 2:** In principle, the same ice crystal habits can be included which are supported by Macke (1994)'s raytracing code. We also included a Python script in the GitHub repository to generate new ice crystal geometries, which currently supports hexagonal prisms with different aspect ratios:
https://github.com/lforster/CrystalTrace/blob/main/crystal_geometry/generate_crystal_file.py

We modified the abstract to clarify this:

*Page 1, line 7: "The current version of CrystalTrace is implemented for the visible spectral range and supports individual hexagonal prisms with different sizes and aspect ratios."*

**Comment 3. Missing diffraction (future work)**
The authors state that diffraction and full-vector (polarized) radiative transfer will be added in a future release. Please estimate how omitting diffraction biases the asymmetry parameter (g) and top-of-atmosphere short-wave flux for thin ($\tau \approx 1$) and thick ($\tau \approx 10$) cirrus layers of different crystal sizes.

**Response 3:** As described in **Response 1**, we included Table 1 with radiative fluxes at TOA and BOA for $\tau \approx 1$ and $\tau \approx 10$ for randomly oriented crystals and compared them with YANG. In contrast to flux, the asymmetry parameters cannot be directly compared between CrystalTrace and YANG. Since our method is based on geometric-optics raytracing, rays that experience only a very small angular deviation are treated as transmission rather than as a scattering event. Consequently, the single-scattering phase function extracted from our radiance field does not include the forward scattering peak which is present in YANG. Our phase function basically corresponds to a delta-scaled representation, with a reduced asymmetry factor, given by g' = (g-f) / (1 -f'). So, the derived g' from CrystalTrace is lower than YANG and a direct comparison would not be meaningful.

**Minor comments:**

**Comment 4:** Please indicate whether CrystalTrace can be configured for active-sensor geometries (e.g. zenith-pointing or spaceborne lidar).

**Response 4**: CrystalTrace takes the zenith and azimuth angle of the incident radiance vector as input and returns the direction of the scattered radiance. In principle, CrystalTrace supports zenith-pointing or spaceborne (nadir-pointing) lidar observation geometries, however simulation with active sensors (lidar/radar) is currently not implemented in MYSTIC.

**Comment 5**: For the benefit of prospective users, consider supplying a concise example input deck (MYSTIC / libRadtran format) that defines an ice cloud with user-selected aspect ratio and orientation parameters

**Response 5**: Thank you for this suggestion. CrystalTrace is part of libRadtran version 2.0.6 and includes an example as test case but has not yet been documented in the user manual. We provide additional documentation and examples in a GitHub repository. We modified the code and data availability statement:

*Page 15, line 261: "CrystalTrace is distributed as part of the libRadtran radiative transfer package (version 2.0.6 and later). Supplementary materials, including detailed documentation, example input files, and plotting scripts used in this work, are available on GitHub (https://github.com/lforster/CrystalTrace) and have been preserved on Zenodo (DOI: 10.5281/zenodo.17679332)."*

**Comment 6**: Include a benchmark case with only randomly oriented crystals, comparing CrystalTrace results to those from an established solver. This will demonstrate that the new code reproduces standard solutions when orientation effects are disabled.

**Response 6**: Figure 7 illustrates the benchmark case using 100% randomly oriented crystals and comparing MYSTIC/CrystalTrace with results from MYSTIC/YANG.

**Comment 7**: Clarify whether the current implementation can handle mixed-phase clouds that contain both ice and liquid water.

**Response 7:** Yes, the current implementation can handle mixed-phase clouds that contain both ice and liquid water. For this use case, two profiles can be defined – one using optical properties of water droplets and the other one for ice crystals. The mixture can be prescribed via their respective optical thickness.

**Specific comments:**

**Comment 8:** p.1 l-14: Please quantify the impact of ice-crystal orientation on the radiation budget (e.g. ΔSW CRE).
**Response 8**: We agree that quantifying the impact of ice crystal orientation on the radiation budget would be valuable. Since this would require a dedicated study with simulations across a large set of parameters (wavelength, crystal shape, orientation fraction, orientation parameter, cloud optical thickness) this is beyond the scope of this study.

**Comment 9:** p.3 l-46: The code is said to build on Macke (1994). Please spell out the improvements and novel aspects relative to that work. Section 2.3 compares CrystalTrace with the bug-fixed version of GO; clarify the differences between the two algorithms.
**Response 9**: We added a more detailed description of the bug-fixed GO version in the Supplement. We wrote a completely new solver in C language, interfacing seamlessly with MYSTIC and libRadtran. For some core raytracing functions, we were drawing inspiration from Macke (1994)'s Fortran code.

**Comment 10:** p.3 l-47: If a broader wavelength range than 400–700 nm is simulated, will the results near 400 nm and 700 nm remain consistent with other algorithms?
**Response 10:** We removed the wavelength interval from this sentence since the effect of the wavelength on the ice crystal size parameter is explained in more details in Section 2. Although we have tested CrystalTrace only in the visible spectrum, it is possible to simulate any other wavelength that is supported by MYSTIC. Since ice absorption is not implemented, spectral variations only lead to changes in the refractive index. The other consideration is that the size parameter of typical ice crystal dimensions is getting smaller with increasing wavelength, deviating more from the Geometric Optics approximation.

**Comment 11:** p.7 l-144: Is photon termination (e.g., Russian-roulette) applied to prevent endless scattering in very thick ice clouds?
**Response 11**: Russian Roulette is implemented in MYSTIC. In the case of ice clouds with very large optical thickness values, the photons will be terminated by MYSTIC before entering CrystalTrace.

**Comment 12:** p.8 l-173: The sentence contains two instances of "with"; please remove one.
**Response 12**: Fixed.

**Comment 13:** p.8 l-184: The authors show that CrystalTrace is unaffected by scattering-angle resolution. What is the computational-cost difference between CrystalTrace and the pre-computed look-up table?
**Response 13:** We do not have access to the source code of Yang et al. 2013 and hence cannot directly compare the total computational cost.

**Comment 14:** p.9 Figure 6: The legends list "4.0e-04" and "2.1e-03", but the meaning of these values is not explained.
**Response 14**: We appreciate the reviewer's suggestion and have updated the caption of Figure 6 accordingly:

*Page 9: "**Figure 6.** Comparison of the scattering [...] Both the GO and CrystalTrace phase functions were scaled to allow for a comparison of the results for scattering angles excluding the forward scattering peak. The scaling factors are shown in the legend. [...]"*

**Comment 15:** p.10 Figure 7c: The background (non-22°) radiance is slightly higher in CrystalTrace. Was ice absorption enabled in the MYSTIC-YANG control run? If not, what explains the difference?

**Response 15**: We appreciate the reviewer's request for clarification. We repeated the simulation and ensured no ice absorption was disabled. The difference of the background radiance between MYSTIC – YANG is of the order of $10^{-3}$. We extended Fig. 7 with the corresponding standard deviation and added the following explanation to the manuscript.

*Page 9, line 167: "Panels (d)–(f) show the standard deviation, representing the Monte Carlo noise within a 1-σ confidence interval, which is on average about 9% relative to the simulated radiances. The mean difference in Fig. 7c is $(0.0004 \pm 0.0573)$ [W m$^{-2}$ nm$^{-1}$ sr$^{-1}$], effectively zero within the 1-σ confidence range."*

**Comment 16:** p.10 l-200: Please provide the standard deviation of radiances for MYSTIC-CrystalTrace versus MYSTIC-YANG over several independent runs.

**Response 16**: We appreciate the suggestion and added results for the standard deviation to Fig. 7.

**Comment 17**: p.10 l-202: Runtime is reported for only $1 \times 10^5$ photons—insufficient for low noise in a hemispheric simulation. Please also give the runtime (and peak memory) for an equivalent MYSTIC-YANG run to substantiate the claimed efficiency.

**Response 17**: Thank you for raising this point. All simulations are performed for $10^7$ photons. We modified the text to be consistent:

*Page 12, line 195: "The runtime for $10^7$ photons amounts to about 50 s for τ = 1 (140 s for τ = 10) using a single core Apple M1 Max processor with 3.2 GHz clock frequency and is the same for CrystalTrace and YANG."*

**Comment 18:** p.13 l-233: How many photons were used to obtain adequate aerosol-scattering statistics?

**Response 18:** The number of photons is the same throughout the study: we used $10^7$ photons. We added the info to the caption of Figure 12:

*Page 14: "**Figure 12.** CrystalTrace simulation of 1 April 2014 8:25 UTC at an SZA of 57° and a wavelength of 500 nm. The simulation was performed for 45% randomly oriented columns which form the 22° halo, 10% oriented plates which cause the sundogs, 35% oriented columns which are responsible for the upper and lower tangent arcs, and 10% Parry-oriented columns which form the upper Parry arc. The simulation was performed with $10^7$ photons. A cirrus optical thickness of 1 was chosen and the aerosol was parameterized by the OPAC desert dust mixture with an AOT of 0.48 at 500 nm, which is the daily mean provided by AERONET."*